# Metabolite Profiling of *Colvillea racemosa* via UPLC-ESI-QTOF-MS Analysis in Correlation to the In Vitro Antioxidant and Cytotoxic Potential against A549 Non-Small Cell Lung Cancer Cell Line

**DOI:** 10.3390/plants13070976

**Published:** 2024-03-28

**Authors:** Álvaro Fernández-Ochoa, Inas Y. Younis, Reem K. Arafa, María de la Luz Cádiz-Gurrea, Francisco Javier Leyva-Jiménez, Antonio Segura Carretero, Engy Mohsen, Fatema R. Saber

**Affiliations:** 1Department of Analytical Chemistry, Faculty of Sciences, University of Granada, Avda Fuentenueva s/n, 18071 Granada, Spain; mluzcadiz@ugr.es (M.d.l.L.C.-G.); javier.leyva@uclm.es (F.J.L.-J.); ansegura@ugr.es (A.S.C.); 2Department of Pharmacognosy, Faculty of Pharmacy, Cairo University, Kasr el-Aini Street, Cairo 11562, Egypt; inas.younis@pharma.cu.edu.eg (I.Y.Y.); engy.mohsen@pharma.cu.edu.eg (E.M.); 3Drug Design and Discovery Lab, Zewail City of Science and Technology, Cairo 12578, Egypt; rkhidr@zewailcity.edu.eg; 4Biomedical Sciences Program, University of Science and Technology, Zewail City of Science and Technology, Cairo 12578, Egypt; 5Department of Analytical Chemistry and Food Science and Technology, University of Castilla-La Mancha, Ronda de Calatrava, 7, 13071 Ciudad Real, Spain

**Keywords:** *Colvillea*, fabaceae, metabolite profiling, mass spectrometry, flavonoid-*C*-glycosides, lung cancer, docking

## Abstract

In this study, flower and leaf extracts of *Colvillea racemosa* were considered a source of bioactive compounds. In this context, the objective of the study focused on investigating the anticancer potential as well as the phytochemical composition of both extracts. The extracts were analyzed by UPLC-ESI-QTOF-MS, and the bioactivity was tested using in vitro antioxidant assays (FRAP, DPPH, and ABTS) in addition to cytotoxic assays on non-small cell lung cancer cell line (A549). Our results clearly indicated the potent radical scavenging capacity of both extracts. Importantly, the flower extract exhibited a greater antioxidant capacity than the leaf extract. In terms of cytotoxic activity, leaf and flower extracts significantly inhibited cell viability with IC_50_ values of 17.0 and 17.2 µg/mL, respectively. The phytochemical characterization enabled the putative annotation of 42 metabolites, such as saccharides, phenolic acids, flavonoids, amino acids, and fatty acids. Among them, the flavonoid *C*-glycosides stand out due to their high relative abundance and previous reports on their anticancer bioactivity. For a better understanding of the bioactive mechanisms, four flavonoids (vitexin, kaempferol-3-*O*-rutinoside, luteolin, and isoorientin) were selected for molecular docking on hallmark protein targets in lung cancer as represented by γ-PI3K, EGFR, and CDK2 through in-silico studies. In these models, kaempferol-3-*O*-rutinoside and vitexin had the highest binding scores on γ-PI3K and CDK2, followed by isoorientin, so they could be highly responsible for the bioactive properties of *C. racemosa* extracts.

## 1. Introduction

Family Fabaceae is a valuable source of many phytochemicals, viz., phenolic acids, flavonoids, saponins, and alkaloids with an anti-cancer potential [1]. Importantly, polyphenolic compounds are primary competitors that can reverse carcinogenesis through inhibition of oxidative stress, induction of cell cycle apoptosis, and suppression of the signaling molecules proliferation [2]. Many studies reported the antioxidant activity of different plants in this family. For instance, Gođevac et al. [3] studied nine Fabaceae species growing in Serbia and Montenegro, where *Lathyrus binatus*, *Trifolium pannonicum*, and *Anthyllis aurea* revealed as the most potent antioxidant plants using different antioxidant assays. Additionally, the seed extract of *Mucuna pruriens* provoked antioxidant effects using DPPH, FRAP, and ABTS assays. The results were comparable to those of rutin and gallic acid used as standards [4]. 

Moreover, the family Fabaceae has been studied for cytotoxic potential. An alkaloidal fraction prepared from *Indigofera suffruticosa* (a member of the family Fabaceae) exhibited cytotoxic activity with IC50 values of 37.39 and 24.13 µg/mL against LM2 (breast adenocarcinoma) and LP07 (lung adenocarcinoma) cell lines, respectively [5]. Also, the extract of *Senna singueana* at 100 mg/kg showed an in vivo antiapoptotic effect on liver cells by reduction of Bcl-2 [6]. Importantly, other studies revealed low cytotoxicity of other plants belonging to Fabaceae on the Vero cell line [7]. 

*Colvillea racemosa* Bojer ex Hook, also known as the whip tree, is an exotic tree that belongs to the Fabaceae family [8]. It represents the only species of the genus *Colvillea*, which is why it is described as a monotypic plant [8]. It is the most beautiful deciduous tree native to the western coast of Madagascar. However, it is widely cultivated as an ornamental plant in some tropical countries such as Australia and North America [8,9]. It is characterized by beautiful, inspiring clusters of bright orange flowers, so it is commonly known as Colville’s Glory. Rivers et al. (2010) [10] reported the morphological similarity between *Colvillea racemosa* and *Delonix regia* or royal poinciana, which are endangered Malagasy species. However, *Colvillea* had a unique genetic material with a greater number of special alleles. In fact, it was placed on the red list as a threatened species according to the International Conservation of Nature and Natural Resources (IUCN). 

Morphologically, the leaves of *C. racemosa* are small compound, bipinnate leaflets with a deep green color [11]. Scientific studies have shown that leaf extracts exhibit multiple pharmacological activities, such as antioxidant, cytotoxic, and antimicrobial activities, in addition to their antineoplastic activity against Ehrlich carcinoma [9,11]. These bioactive properties are associated with the phytochemical composition of this matrix, which has been characterized mainly by the presence of phenolic compounds (e.g., hesperidin, naringin, vanillic and benzoic acids) [11] and dihydrochalcones [8]. However, only a very small number of studies have focused on the phytochemical characterization of *C. racemosa* extracts. Therefore, there is a need to explore in detail the phytochemical compounds present in this species, which can help to better understand the bioactive properties described. 

Furthermore, several classes of chemical compounds, such as flavonoids, diterpenes, and steroids, have been isolated from the closely related genus *Caesalpinia* [12]. Members of this genus are endowed with an unusual combination of biologically active carotenoids and anthocyanins, particularly cyanidin-3-*O*-rutinoside, which holds great promise in food and pharmaceutical applications [13]. 

Lung carcinoma remains the first leading cause of death worldwide and has a low survival rate, according to recent reports from the World Health Organization. In 2020, it was responsible for more than one million deaths, accounting for approximately 18% of all cancer deaths [14]. Lung cancer can be divided into small-cell lung cancer (SCLC) and non-small cell lung cancer (NSCLC) groups [15]. NSCLC constitutes approximately 85% of all lung cancer cases [16,17]. Notably, surgery is the preferred and main approach for early-stage NSCLC, which offers a significant chance for long-term survival for those patients [18,19]. However, late stages require platinum-based combinations as chemotherapeutics [20]. 

Chemotherapy and radiotherapy are the most common conventional therapies for cancer treatment. However, they suffer from drug resistance and serious side effects [21]. Therefore, as a potential future alternative, plant-based secondary metabolites show promising anticancer activities with multiple targets. 

Importantly, oxidative stress stimulates the signaling pathway phosphatidylinositol 3-kinase (PI3K)/AKT, which in turn triggers apoptosis [22]. Accordingly, PI3K and other kinases have emerged as attractive targets for the therapeutic intervention of cancer. PI3K plays a crucial role in the DNA repair cell cycle, cellular metabolism, and programmed cell death [23]. Similarly, the epidermal growth factor receptor (EGFR) is another type of kinase that plays a key role in cell growth, differentiation, and tumorigenesis. A previous study by Filosto et al. [24] demonstrated that oxidative stress induced aberrant phosphorylation of the EGFR in the A549 adenocarcinoma model. Therefore, targeting these two pathways is considered 0a promising therapeutic strategy for a variety of malignancies such as stomach, breast, colon, and lung [23,25]. 

Another key player in cancer pathogenesis is CDK2, which is a serine/threonine protein kinase that plays a role in the G1/S transition, the initiation of DNA synthesis, and the regulation of the exit from the S phase. CDK2 is altered in 0.16% of all cancers, yet specifically colon adenocarcinoma possesses the greatest prevalence of alterations [26].

Recently, there has been a growing interest in natural antioxidants that can capture reactive oxygen species (ROS) production, reduce oxidative damage, induce DNA damage, and decrease the generation of anti-apoptotic protein levels [27]. In this context, there is still a great need to investigate the mechanisms of action of these plant-based secondary metabolites, which may lead to possible pharmacological alternatives for cancer treatment in the future.

To the best of our knowledge, only a few studies on *Colvillea* have been conducted. Therefore, our study aimed to comprehensively investigate the bioactive metabolites of *C. racemosa* leaf and flower extracts using UPLC-ESI-QTOF-MS and their antioxidant potential as well as their cytotoxic effects targeting A549 non-small cell lung carcinoma (NSCLC) by in-vitro assays and molecular modeling approaches. We anticipate that this study will provide guidelines for the development, utilization, and pharmaceutical application of *C. racemosa*.

## 2. Results and Discussion

### 2.1. Total Phenolic and Flavonoid Contents

As depicted in Table 1, both phenolic and flavonoid contents of the flower extract of *C. racemosa* were approximately three times higher than that of the leaf extract (98.24 and 86.68 versus 34.99 and 24.14, for total phenolics and flavonoids calculated as mg gallic acid/g extract and rutin equivalents/g extract, respectively). 

### 2.2. In-Vitro Antioxidant Assays

Radical scavenging capacity is essential for suppressing the harmful effects of free radicals on the human body. Leaf and flower extracts of *C. racemosa* were evaluated for their antioxidant potential using three different models of antioxidant assays, viz., FRAP, DPPH, and ABTS assays. The results clearly indicated the antioxidant activities of both extracts (Table 1). However, the flower extract exhibited a significantly higher antioxidant capacity (IC_50_: 190 ± 8.0 µM TE/mg extract) for reducing Fe^3+^ to Fe^2+^ in an acidic medium in the FRAP assay compared to the leaf extract (IC_50_: 362 ± 24 µM TE/mg extract). Similarly, the flower extract showed a more potent radical scavenging capacity (IC_50_: 177 ± 2.0, 26 ± 2.0 µg/mL) than the leaf extract (IC_50_: 381 ± 2.0, 36 ± 2.0 µg/mL) using DPPH and ABTS methods, respectively. The unique diversity of flavonoid structures is what ultimately results in a marked difference in biological activities. In fact, flavonoids (kaempferol, luteolin, and apigenin), as well as non-flavonoid compounds such as phenolic acids, act as natural radical scavenging agents and metal chelators since they can neutralize free radicals via electron donation [28].

### 2.3. UPLC-ESI-QTOF-MS Metabolite Profiling of C. racemosa

After data processing, 49 potential molecular features were deconvoluted and annotated. Information on these molecular features (RT, *m*/*z*, formula molecular, relative abundances) is shown in Table 2. According to the identification guidelines proposed by Sumner et al. (2007) [29], 29 compounds were annotated based on their MS/MS spectra and compared with those spectra present in databases (level 2), 13 solely based on their molecular formula and MS^1^ spectra (level 3), and 7 molecular features remained unknown (level 4). Different saccharides, amino acids, organic acids, fatty acids, phenolic compounds, and flavonoids were annotated in the analyzed extracts. LC/MS chromatograms of the methanolic extracts of *C. racemosa* leaves and flowers are depicted in Figure 1.

#### 2.3.1. Phenolic Acids

Phenolic acids are among the simplest secondary metabolites formed of one phenolic ring with multiple hydroxy or methoxy groups and display high potential for use as anticancer agents with diverse targets [39]. Benzoic acid derivatives, either as aglycones or glycosidic conjugates, were the most abundant phenolic acids identified in the flowers and leaves of *C. racemosa*. Protocatechuic acid and dihydroxybenzoic acid pentoside displayed [M − H]^−^ at *m*/*z* 153.0193 (C_7_H_5_O_4_^−^) and 285.0615 (C_12_H_13_O_8_^−^) with a characteristic loss of CO_2_ moiety [31]. Protocatechuic and gallic acids were also previously reported in a characterization study of *Colvillea* based on HPLC-DAD [11]. Notably, protocatechuic acid exerted antiproliferative effects on multiple cancer cell lines, viz., HL-60 leukemia, gastric adenocarcinoma, ovarian carcinoma A2780, and MCF-7 [39]. 

#### 2.3.2. Flavonoids

Flavonoids are the most important polyphenolic secondary metabolites with diverse structures and functions. Based on MS/MS analysis, approximately 18 flavonoids belonging to different subclasses, viz., flavone-*C*-glycosides, flavones, flavonols, and isoflavones, have been identified. These compounds are presented mainly as glycosidic conjugates (Table 2). The linkage and the nature of sugars on the flavonoid skeleton occur either by attachment to the hydroxyl group, as in *O*-glycosides, or by direct linkage to the carbon atom in the A-ring, as in *C*-glycosides [40]. 

#### 2.3.3. Flavone-C-Glycosides

In *C. racemosa*, luteolin and apigenin conjugates represented the unique landmarks of the plant. They were detected as mono or di-glycosides where the sugar substitution is typically present at C-6 and/or C-8. The key fragmentation patterns of apigenin glycosides showed the sequential loss of 120 and 90 amu for hexose and 90 and 60 amu for pentosyl residue, respectively. Peaks 18 and 20 displayed identical [M − H]^−^ signals at *m*/*z* 431 (C_21_H_19_O_10_^−^) with congruent product ions at *m*/*z* 283 [M − H − 120]^−^ and 341 [M − H − 90]^−^. Nevertheless, they appeared at different retention times (Rt 26.42 min and 28.03 min) and were tentatively annotated as apigenin-8-*C*-glucoside (vitexin) and apigenin-6-*C*-glucoside (iso-vitexin), respectively [33]. These results are in good agreement with those of Mohamed et al. (2018) [8], who isolated both compounds from the stems of *C. racemosa*. To the best of our knowledge, rhoifolin, luteolin 6-*C*-hexoside-8-*C*-arabinoside, apigenin-6-*C*-hexose-8-*C*-pentoside I, and apigenin-6-*C*-hexoside-8-*C*-pentoside II were identified for the first time in *C. racemosa*. Apparently, apigenin glycosides were previously reported to significantly inhibit cell proliferation and suppress lung cancer through downregulation of the IFN-γ signaling pathway [41,42]. 

#### 2.3.4. Flavonols

Unlike *C*-flavonoids, most flavonoid-*O*-glycosides could be simply assigned by the neutral loss of sugar moieties, such as 162 amu for hexose, 146 amu for deoxyhexose, and 132 amu for pentose, resulting in the formation of their corresponding aglycones [40]. Kaempferol-3-*O*-rutinoside (peak 26) was the most prominent flavonol identified at *m*/*z* 593.1511 (C_27_H_29_O_15_^−^), which showed a characteristic loss of rutinoside unit to yield an intense peak at *m*/*z* 285 [37]. Likewise, kaempferitrin (kaempferol di-rhamnoside) (peak 28) characteristically exhibited the molecular ion at *m*/*z* 577.1562 (C_27_H_29_O_14_^−^) and showed its respective aglycone after one-step cleavage of two rhamnose molecules [36]. Despite the previous identification of kaempferol aglycone in the stem and leaves of *C. racemosa* [8,11], the latter two glycosides were putatively identified for the first time in *C. racemosa*. Recently, kaempferol significantly inhibited the proliferation of lung cancer A549 cells through the downregulation of claudin-2 expression [43]. Moreover, limocitrin (peak 29), a unique *Citrus* flavonol, was tentatively assigned based on its indicative fragment ions at *m*/*z* 345 and 330 [37].

#### 2.3.5. Isoflavones

Intriguingly, daidzein (peak 30) was the only isoflavone tentatively identified for the first time in *Colvillea*. It has been shown to play an important role in the regulation of cancer-related signaling pathways, especially in ovarian cancer [41].

#### 2.3.6. Fatty Acids

Structurally, fatty acids (FAs) are long-chain hydrocarbons with reactive carboxylic acid groups. These compounds exhibited a potential broad-spectrum activity against cancer metabolism and proliferation [44]. Oleic and linoleic acids (peaks 36 and 38) were among the most abundant fatty acids detected at the end of the chromatographic run at *m*/*z* 281.2486 and 279.2329 with molecular formulas (C_18_H_33_O_2_^−^) and (C_18_H_31_O_2_^−^), respectively. 

### 2.4. In Vitro Cytotoxicity against A549 Cell Line and Influence of C. racemosa Extracts on Cell Cycle Distribution

To assess the cytotoxic potential of the *C. racemosa* extracts, an SRB cytotoxicity assay was performed against the lung cancer cell line (A549). The results are shown in Figure 2. Both leaves and flower extracts inhibited the cell viability with IC_50_ values of 17.0 and 17.2 µg/mL, respectively. Accordingly, the effect of leaf and flower methanolic extracts of C. racemosa on cell cycle distribution was assessed in the same cell line using flow cytometry, and the results are presented in Figure 3. The flower extract of C. racemosa displayed an increase in the cells in the S phase compared to the control (26.9 ± 1.0% to 27.9 ± 1.3%), suggesting S phase cell cycle arrest [45]. Furthermore, a significant increase in the G2/M phase was also evoked by *C. racemosa* flower extract (15.2 ± 0.9% to 19.0 ± 1.6%). Interestingly, both leaf and flower extracts of *C. racemosa* showed a significant increase in the cells in the pre-G phase from 13.0 ± 0.2% to 15.3 ± 0.5% and 17.3 ± 0.3%, compared to the control group, respectively. 

Previous studies highlighted the potential of flavonoid-C-glycosides as cytotoxic agents that affect cancer cell cycle distribution. A mechanistic study by Zhang et al., 2018) subsequently demonstrated that vitexin caused G2/M cell cycle arrest and, consequently, an accumulation of cell population was observed in this phase. This effect also caused G0/G1 to significantly decrease after treatment of the human glioblastoma cell line LN-18 with vitexin at the micromolar level [46]. In this context, Yang and co-workers (2018) [47] reported that the cell cycle arrest at the G2/M phase was demonstrated by vicenin-2, also with a concurrent significant decrease in the G0/G1 phase. Importantly, the O-glycoside kaempferol rutinoside effectively induces intrinsic apoptosis in lung adenocarcinoma through mitochondrial dysfunction and calcium overload [48]. 

Notably, isoorientin was reported to induce G2/M cell cycle arrest in the A549 cell line. This effect, in addition to its apoptotic activity, was mediated via the signaling pathway of MAPK/STAT3/NF-κB, which supports its potential as a lead drug in lung cancer management [49]. 

Nevertheless, orientin triggered cell cycle arrest at the G0/G1 phase in the HT-29 human colorectal cancer cell line [50]. These findings could further support the results of metabolic profiling of *C. racemosa* extracts with vitexin, isoorientin, and vicenin-2 as the major detected flavonoid-C-glycosides, in addition to kaempferol rutinoside. Thus, the antiproliferative effect of Colvillea flower extract could be mediated by these major flavonoid glycosides, mostly affecting the G2/M phase. Noteworthy, the increase in the dead cell population (pre-G phase) is indicative of cell death induced by the encompassed bioactive compounds of *C. racemosa*.

### 2.5. Docking Results

Molecular docking is an important computational technique to predict the best interaction between a ligand and its functional site [51,52]. Phosphoinositide 3-kinases (PI3Ks) are a unique group of lipid kinases that represent the key regulators of many cellular functions, such as growth, metabolism, proliferation, differentiation, invasion, and immune signaling [23]. Therefore, PI3K signaling is extensively studied as a promising target for combating cancer. Unambiguously, plant-based polyphenolic compounds have been reported to act as natural anticancer agents through a myriad of mechanisms, viz., inhibition of reactive oxygen species (ROS), activation of apoptosis, and suppression of protein kinases [14,53]. Inspired by this concept, the most abundant flavonoids, viz., vitexin, kaempferol rutinoside, luteolin, and isoorientin, were analyzed on γ-PI3K through an in silico study, and the docking interactions are shown in Figure 4. As illustrated in Table 3, kaempferol rutinoside and vitexin showed the best binding scores among all investigated flavonoid compounds (−8.7140 and −7.5713 kcal/mol, respectively) compared to the docked co-crystallized inhibitor QYT (−6.6005 kcal/mol). They were stabilized into PI3K pocket through the formation of H-bonds with the amino acids Asp 964, Asp 950, Val 882, Ile 879, Ser 806, Thr 887, and Lys 890, rationalizing their potential activity as anticancer agents. Noteworthy, the docking results of kaempferol rutinoside are in line with those of Li et al. [48], who reported that kaempferol rutinoside, the unique flavone of Tetrastigma hemsleyanum successfully inhibited the cellular viability in A549 in a dose-dependent manner. 

Additionally, EGFR is another important target in cancer management for cancers associated with the overexpression of this receptor tyrosine kinase, like colorectal cancer. This prompted us to investigate the binding interaction potential of the most abundant flavonoids, viz., vitexin, kaempferol rutinoside, luteolin, and isoorientin with EGFR. Table 3 and Figure 5 display the docking findings of the examined compounds in comparison to the co-crystallized ligand AQ4 (S-score −8.5478 kcal/mol). Results showed that all four compounds demonstrated a good recognition of the active site amino acids with S-scores of −7.3955, −9.3078, −6.2791, and −8.0587 kcal/mol, respectively. Kaempferol rutinoside showed better binding than the co-crystallized ligand as reflected by its S-score forming a network of H-bonds with Ala 719, Asp 831, Arg 817, and Asn 818 active site amino acids.

Furthermore, CDK2 is also considered an important protein target in cancer [26]. As such, we wanted to determine the potential effect of the selected flavonoids, viz., vitexin, kaempferol rutinoside, luteolin, and isoorientin on CDK2. In silico investigation, results reflected the ability of these flavonoids to recognize and interact with the key amino acids of the active site of CDK2 in a comparable manner to the co-crystallized inhibitor DTQ (Table 3 and Figure 6). While the co-crystallized ligand displayed an S-score for binding of −7.1365 kcal/mol, the investigated flavonoids had better or near S-scores of −7.5700, −8.5236, −6.3872, and −6.8739 kcal/mol, respectively. The best binder again was kaempferol rutinoside, forming two H-bonds with both Asp 145 and Val 18, in addition to a pi-H interaction with Ala 144. 

Details of the 3D interactions of the most active compound in the docking study by virtue of its high binding S-scores, kaempferol rutinoside, are presented in Figure 7. With all three biological targets, PI3K, EGFR, and CDK2, kaempferol rutinoside was able to establish a network of H-bonds and pi-H bonding interactions with the active site amino acids to stabilize its complex with these biotargets and hence elicit its inhibitory activity paving the way to the anticancer effect of *C. racemosa* it participates to. 

Ultimately, our docking study provides scientific evidence for the promising anticancer activity of *C. racemosa*.

On the other hand, aside from our in vitro and in silico findings on the cytotoxic potential of *C. racemosa* extracts, extrapolation on in vivo testing will be our future perspective. Also, detailed risk assessment and systemic toxicity profile should be investigated to guard against any cytotoxic effects on normal tissues. Additionally, after sufficient in vivo exploration, the synergistic effect of *C. racemosa* extracts with drugs used for NSCLC will be highly valuable for decreasing the side effects of these synthetic drugs.

## 3. Materials and Methods

### 3.1. Plant Material 

Leaves and flowers of *C. racemosa* Bojer ex Hook were collected in July 2019 from Mostafa EL-Abd Gardens, Cairo-Alexandria Desert Road, Egypt (30.3552330 N, 30.5166490 W). The plant material was kindly identified by Mrs. Therese Labib, Consultant of Plant Taxonomy at the Ministry of Agriculture and Orman Botanical Garden, Giza, Egypt. Voucher specimen (No. 10.7.2019) was kept in the herbarium of the Department of Pharmacognosy, Faculty of Pharmacy, Cairo University. Leaves and flowers were air-dried in the shade, powdered, and stored in airtight containers until subsequent analysis. A stock solution of each *C. racemosa* extract (5 mg/mL) was prepared for UPLC-ESI-QTOF-MS analysis.

### 3.2. UPLC-ESI-QTOF-MS Metabolite Profiling of Leaves and Flowers of C. racemosa

Flower and leaf extracts of *C. racemosa* were analyzed using an ACQUITY UPLC H-Class System (Waters, Milford, MA, USA) coupled to a QTOF-MS (Synapt G2, Waters Corp., Milford, MA, USA). A reversed-phase C18 column (Agilent Zorbax Eclipse Plus, 2.1 × 150 mm, 3.5 μm) was used for the chromatographic separation. The mobile phases were acidified water with acetic acid at 0.5% (phase A) and methanol (phase B). The following mobile phase gradient was established for optimal chromatographic separation: 0 min (100% A), 5 min (75% A), 20 min (40% A), 38 min (0% A), and 46–56 min (100%A). The flow rate, column temperature, and injection volume were set at 400 µL/min, 22 °C, and 10 µL, respectively. Mass detection was performed over a range of mass to charge ratio (*m*/*z*) from 50 to 1200, using negative electrospray ionization mode (ESI-). MS acquisition was carried out using two scan functions by fast switching, in which one of them was operated with a low collision energy in the gas cell (4 eV) and the other with a high collision energy using a ramp from 20 to 60 eV (MSE mode). Leu-Enkephalin was continuously injected for mass calibration. Other MS parameters were as follows: desolvation temperature 500 °C; source temperature 100 °C; desolvation gas flow 700 L/h; cone gas flow 50 L/H; scan; capillary voltage 2.2 kV; cone voltage 30 V; duration 0.1 s; and resolution 20,000 FWHM. 

### 3.3. MS Data Processing 

The raw data files were first converted to .mzML format using MSConver GUI software (Proteowizard; https://proteowizard.sourceforge.io/ (accessed on 15 January 2024)) [54]. The MS data processing steps were carried out using MZmine 2.53 [55]. The following parameters were set for the different stages. (1) Noise level: 1 × 10^3^. (2) ADAP chromatogram builder method: min number of scans, 9; *m*/*z* tolerance, 10 ppm; group intensity threshold, 1 × 10^3^ min highest intensity, 1 × 10^4^. (3) Chromatogram deconvolution—wavelets (ADAP): S/N threshold, 50; coefficient/area threshold, 110; min feature height, 5 × 10^4^; duration range, 0.05–0.3 min; RT wavelet range, 0–0.30. (4) Isotopic peak grouper: RT tolerance, 0.02 min; *m*/*z* tolerance, 10 ppm. (5) Alignment—Join Aligner method: *m*/*z* tolerance, 10 ppm; RT tolerance, 0.1 min; maximum charge, 2. The signals which were also deconvoluted in the analytical blanks were filtered out. The annotation step was performed by comparing the MS fragments acquired in the MSE scans with different metabolomics databases (FoodDB, HMDB, Kegg, etc.), MassBanks (MONA, MassBank Europe Mass Spectral DataBase), and Sirius 4.4.29 [56].

### 3.4. Measurements of Total Phenolics and Flavonoids and In-Vitro Antioxidant Assays

#### 3.4.1. Total Phenolics and Flavonoids

Stock solutions of gallic acid and rutin in methanol (1 mg/mL) were prepared, and seven serial dilutions were acquired in the concentrations of (50, 100, 200, 400, 600, 800, 1000) µg/mL, and standard calibration curves were constructed accordingly. The leaf and flower extracts of *C. racemosa* were prepared at concentrations of 4 mg/mL in methanol. Each of the standards and two plant extracts were pipetted into the plate wells in six replicates. Measurements were performed at 630 and 420 nm for phenolics and flavonoids, respectively, using a FluoStar Omega microplate reader (BMG LABTECH, Ortenberg, Germany) [57,58]. 

#### 3.4.2. FRAP Assay

The FRAP assay was established according to the standard Fe^3+^ reducing power assay of Benzie and Strain [59] with some modifications to be suitable for microplates. Briefly, solutions of leaves and flowers of *C. racemosa* were prepared in a concentration of 1 mg/mL in methanol. Similarly, Trolox stock solutions (10 mg/mL) in methanol with serial dilutions were prepared at concentrations of 500, 250, 125, 62.5, 31.2, 15.6, and 7.8 µg/mL. The 2,4,6-tri(2-pyridyl)-s-triazine (TPTZ) reagent was freshly prepared (300 mM acetate buffer (pH = 3.6), 10 mM TBTZ in 40 mM HCl, and 20 mM FeCl_3_, in a ratio of 10:1:1 *v*/*v*/*v*, respectively. TPTZ (190 µL) was mixed with 10 µL of the investigated *Colvillea* extracts in a 96-well plate (n = 6), and the reaction mixture was incubated at room temperature (RT) for 30 min in darkness. Finally, the absorbance of triplicate samples was measured at 593 nm using a FluoStar Omega microplate reader (BMG LABTECH, Ortenberg, Germany). The ferric-reducing ability is presented as µM TE/ mg dried extract. 

#### 3.4.3. DPPH Assay

A DPPH (2,2-diphenyl-1-picryl-hydrazyl-hydrate) free radical assay was performed to evaluate the free radical scavenging potential of the *Colvillea* extracts as previously described by Boly et al. (2016) [60]. Briefly, solutions of the leaves and flowers of *C. racemosa* (1 mg/mL) were serially diluted to provide five concentrations (400, 300, 200, 100, and 50 µg/mL). DPPH reagent 100 µL (0.1%) was added to 100 µL of the extract in a 96-well plate (n = 6). The reaction mixture was incubated at room temperature for 30 min under darkness. After incubation, the absorbance was measured at 540 nm. Each sample was measured in triplicate. The results were acquired using a FluoStar Omega microplate reader. The data were analyzed using Microsoft Excel 2016^®^, and the IC_50_ value was calculated using GraphPad Prism 8^®^ [61]. 

#### 3.4.4. ABTS Assay

The ABTS assay was carried out as previously described by Arnao et al. (2001) [62], with some modifications. Briefly, several concentrations of leaves and flowers of *C. racemosa* were prepared (100, 200, 400, 600, and 800 µg/mL) from a stock solution of 1mg/mL in methanol. One milliliter of ABTS solution (192 mg/50 mL distilled water) was added to 17 µL of 140 mM potassium persulfate and left in the dark for 24 h. Then, 1 mL of the reaction mixture was diluted to 50 mL, and 190 µL of the freshly prepared ABTS reagent was mixed with 10 µL of *C. racemosa* extract in a 96-well plate (n = 6). The reaction mixture was incubated at room temperature for 120 min in the dark. The absorbance was measured at 734 nm.

### 3.5. In-Vitro Cytotoxic Activity of C. racemosa on Lung Cancer Cell Lines

#### 3.5.1. Cytotoxicity Assay 

The sulforhodamine B (SRB) assay was a well-established method for screening the cytotoxicity of plant extracts. In brief, different concentrations (0.01, 0.1, 1, 10, and 100 µg/mL) of both leaf and flower extracts from *C. racemosa* were used. Lung cancer cell lines (A-549) were obtained from Nawah Scientific Inc. (Cairo, Egypt). The cells were maintained in DMEM media supplemented with 100 mg/mL of streptomycin, 100 units/mL of penicillin, and 10% of heat-inactivated fetal bovine serum in a humidified, 5% (*v*/*v*) CO_2_ atmosphere at 37 °C. Cell viability was evaluated by SRB assay [63,64]. Aliquots of 100 μL cell suspension (5 × 10^3^ cells) were incubated in 96-well plates for 24 h. Then, the suspension was treated with 100 μL media containing the extracts at various concentrations ranging from (0.01, 0.1, 1, 10, and 100 µg/mL) for 72 h. Fixation of the cells was performed by replacing the media with 150 μL of 10% TCA and incubated at 4 °C for 1 h. Finally, the TCA solution was removed, and the cells were washed with distilled water, followed by the addition of 70 μL of SRB solution (0.4% w/v) and incubated at room temperature for 10 min in the dark. The plates were washed with 1% acetic acid (3 times) and left to dry overnight. Then, 150 μL of 10 mM Tris was added to solublize protein-bound SRB stain, and the absorbance was measured at 540 nm by a BMG LABTECH^®^-FLUOstar Omega microplate reader (Ortenberg, Germany). 

#### 3.5.2. Analysis of Cell Cycle Distribution

The cells were treated with leaf and flower extracts of *C. racemosa* for 48 h. Cells (105 cells) were collected by trypsinization and then washed twice with ice-cold PBS (pH 7.4). The cells were re-suspended in 60% ice-cold ethanol (2 mL) and incubated at 4 °C for 1 h to allow fixation. Fixed cells were washed with PBS (pH 7.4) twice and suspended in 1 mL of PBS containing 50 µg/mL of RNAase A and 10 µg/mL of propidium iodide. After 20 min, the DNA contents of the cells were analyzed via flow cytometry analysis using an FL2 (λex/em 535/617 nm) signal detector (ACEA Novocyte™ flow cytometer, ACEA Biosciences Inc., San Diego, CA, USA). For each sample, 12,000 events were recorded. Cell cycle distribution was calculated using the software of ACEA NovoExpress™ version 1.1.0 (ACEA Biosciences Inc., San Diego, CA, USA) [45,65].

### 3.6. Molecular Docking 

The binding modes of selected compounds and their binding sites were established using Molecular Operating Environment software (MOE, 2019.01). All minimizations were performed with MOE until an RMSD gradient of 0.1 kcal∙mol^−1^Å^−1^ with MMFF 94x force field, and the partial charges were automatically calculated. The X-ray crystallographic structure of γ-PI3K enzyme co-crystalized with 5-quinoxalin-6-ylmethylene-1,3-thiazolidine-2,4-dione (QYT) (PDB ID: 2A5U), EGFR co-crystallized with [6,7-bis(2-methoxy-ethoxy)quinazoline-4-yl]-(3-ethynylphenyl)amine (AQ4) (PDB ID: 1M17), and CDK2 co-crystallized with 4-[3-hydroxyanilino]-6,7-dimethoxyquinazoline (DTQ) (PDB ID: 1DI8), retrieved from Protein Data Bank (https://www.rcsb.org/ (accessed on 15 January 2024)). Four flavonoids were selected for docking, and their structure was retrieved from the PubChem database (http://pubchem.ncbi.nlm.nih.gov (accessed on 15 January 2024)). For each co-crystallized enzyme, water molecules and ligands that are not involved in the binding were removed, and the protein was prepared for the docking study using the Protonate 3D protocol in MOE with default options. The co-crystalized ligand (QYT) was used to define the binding site for docking. The Triangle Matcher placement method and London dG scoring function were used for docking.

### 3.7. Statistical Analysis

The results of antioxidant, in-vitro cytotoxicity, and cell cycle analyses are presented as the mean ± SEM using GraphPad Prism 8 software. One-way ANOVA followed by Tukey’s post hoc test was used to test the significance; *p* < 0.05 was taken as the cut-off value for significance.

## 4. Conclusions

The current study describes the detailed metabolic profiling of flower and leaf extracts of *Colvillea racemosa*, which resulted in the putative annotation of 42 metabolites, including saccharides, phenolic acids, flavonoids, amino acids, and fatty acids. The antioxidant activity and cytotoxicity against non-small cell lung cancer cell lines were demonstrated in this study for both extracts, highlighting their cytotoxic potential. The *C*-glycoside flavonoids characterized in both extracts stand out due to their high abundance and the properties previously reported for this class of compounds. The docking results of four major flavonoids showed that kaempferol-3-*O*-rutinoside and vitexin had the highest binding scores on γ-PI3K and CDK2, followed by isoorientin. Meanwhile, kaempferol-3-*O*-rutinoside and isoorientin exhibited the best binding scores on EGFR. Thus, our docking results rationalize the potential in vitro activity of *C. racemosa* extracts as cytotoxic agents against NSCLC. Despite these promising results, more studies are needed to isolate these potentially active ingredients and to reveal the corresponding extensive potential mechanism of action, which allows its use in possible future therapeutic applications.

## Figures and Tables

**Figure 1 plants-13-00976-f001:**
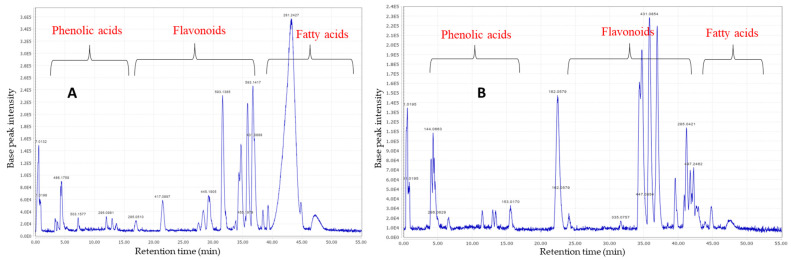
LC/MS base peak chromatograms of methanolic extracts of *C. racemosa* leaves (**A**) and flowers (**B**).

**Figure 2 plants-13-00976-f002:**
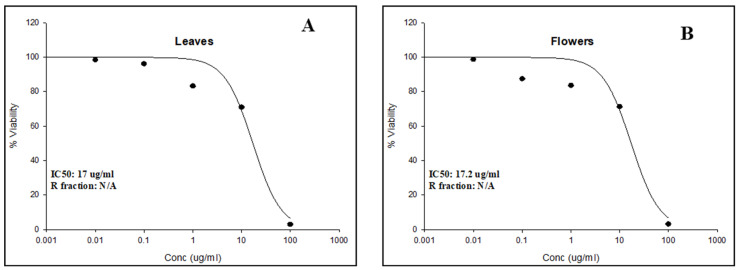
In vitro cytotoxic potential of leaf (**A**) and flower (**B**) extracts of *Colvillea racemosa* against A549 lung cancer cell line (IC_50_ values are shown in the graphs).

**Figure 3 plants-13-00976-f003:**
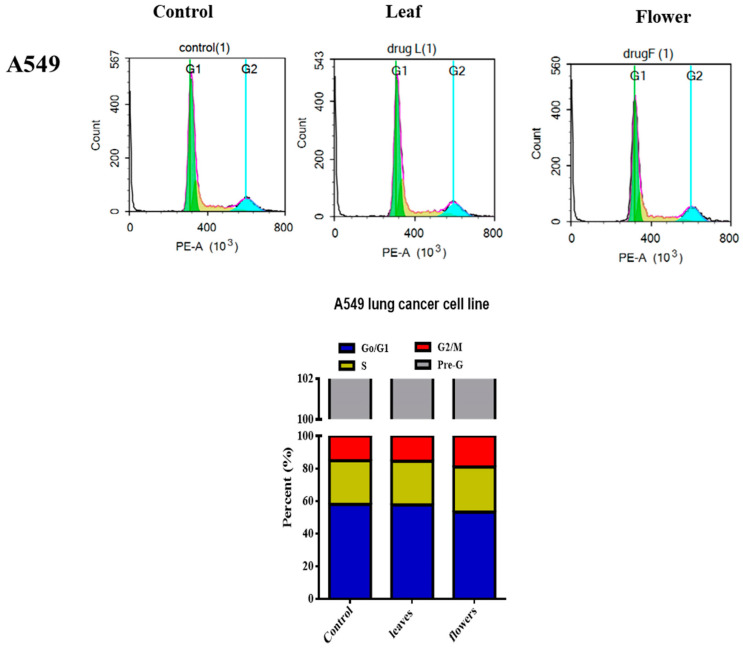
Diagrammatic representation of cell cycle analysis using flow cytometry following 24 h incubation of leaf and flower extracts of *Colvillea racemosa* at their IC_50_ with A549 lung cancer cell line.

**Figure 4 plants-13-00976-f004:**
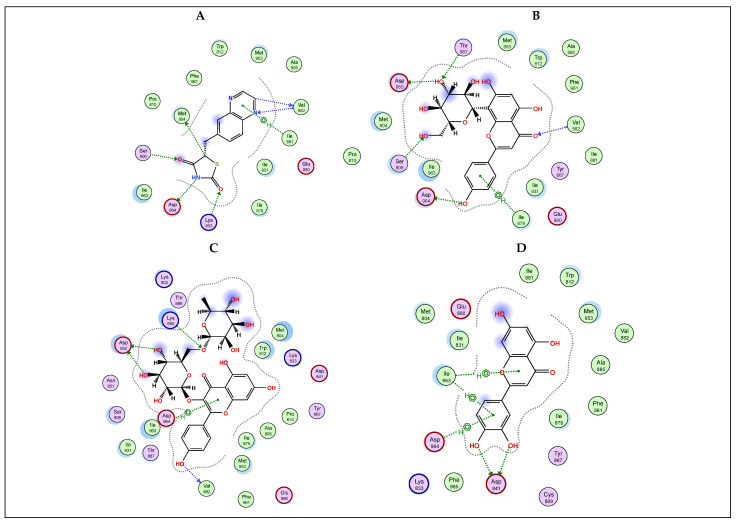
2D docking interaction diagrams showing the pose of the standard drug QYT (**A**), selected *Colvillea* phytochemicals of vitexin (**B**), kaempferol-3-*O*-rutinoside (**C**), luteolin (**D**), and isoorientin (**E**) with the key amino acids in γ PI3K binding site.

**Figure 5 plants-13-00976-f005:**
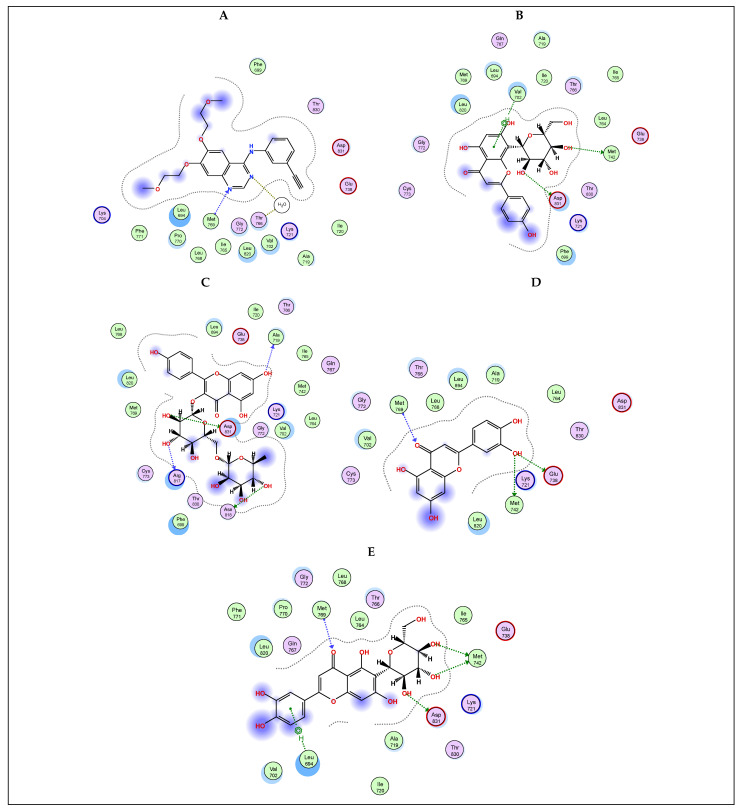
2D docking interaction diagrams showing the pose of the standard drug AQ4 (**A**), selected *Colvillea* phytochemicals of vitexin (**B**), kaempferol-3-*O*-rutinoside (**C**), luteolin (**D**), and isoorientin (**E**) with the key amino acids in EGFR binding site.

**Figure 6 plants-13-00976-f006:**
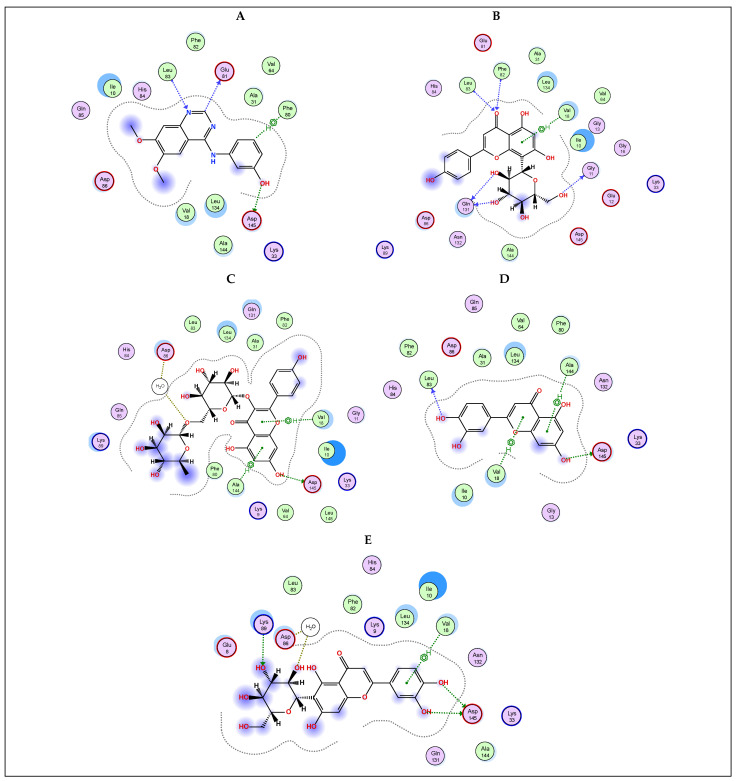
2D docking interaction diagrams showing the pose of the standard drug DTQ (**A**), selected *Colvillea* phytochemicals of vitexin (**B**), kaempferol-3-*O*-rutinoside (**C**), luteolin (**D**), and isoorientin (**E**) with the key amino acids in CDK2 binding site.

**Figure 7 plants-13-00976-f007:**
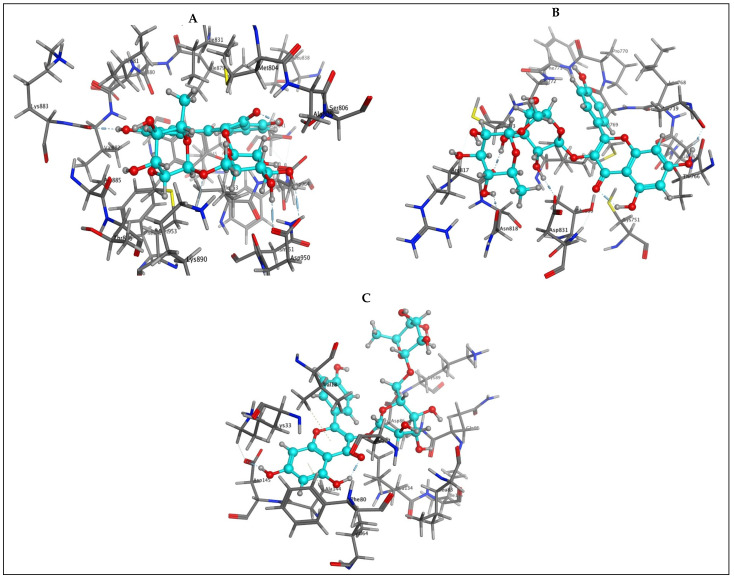
3D docking interaction diagrams showing the pose and interactions of kaempferol-3-*O*-rutinoside (cyan carbons) with PI3K (**A**), EGFR (**B**), and CDK2 (**C**) key active site amino acids.

**Table 1 plants-13-00976-t001:** Quantitative determination of total phenolic and flavonoid contents and in-vitro antioxidant assays of leaves and flowers of *C. racemosa* extracts.

Extract	Total Phenolic Content (mg Gallic acid/g Extract) **#**	Total Flavonoid Content (mg Rutin/g Extract) **#**	FRAP (µM TE/mg Extract) **#**	DPPH IC_50_ (µg/mL) *****	ABTS IC_50_ (µg/mL) *****
Leaves	34.99 ± 3.50 ^a^	24.14 ± 1.60 ^a^	362.05 ± 23.64 ^b^	381.0 ± 2.0 ^c^	36.28 ± 1.82 ^c^
Flowers	98.24 ± 3.40 ^b^	86.68 ± 3.48 ^b^	190.40 ± 8.29 ^a^	177.0 ± 2.0 ^b^	26.08 ± 1.56 ^b^
Trolox	---	---	---	9.77 ± 0.53 ^a^	2.82 ± 0.10 ^a^

a, b, c, Different letters denote significant differences between treatments using # unpaired *t*-test and * one-way ANOVA followed by Tukey’s at *p* < 0.05.

**Table 2 plants-13-00976-t002:** Metabolite profiling of the methanolic extracts of leaves and flowers of *C. racemosa* as analyzed by UPLC-ESI-QTOF-MS.

Nº	RT	*m*/*z* exp. [M − H]^−^	*m*/*z* Theoretical [M − H]^−^	Error (ppm)	Neutral Formula	Compound	MS/MS	Leaf ^1^	Flower ^1^	Reference
**Sugars**										
**1**	4.02	195.0497	195.0510	−6.66	C_6_H_12_O_7_	Gluconic acid	75(77), 129(54), 195(100)		+	HMDB00625
**2**	4.30	144.0666	144.0666	0.00	C_6_H_11_NO_3_	4-Acetamido-butanoate	Level 3	+	+	HMDB03681
**3**	4.48	341.1078	341.1089	−3.22	C_12_H_22_O_11_	Disaccharide-trehalose	59(7), 71(17), 89(25), 101(28), 179(47), 341(100)	+	+	[30]
**4**	4.49	503.1617	503.1617	0.00	C_18_H_32_O_16_	Raffinose	161(7), 179(14), 221(3), 323(15), 341(29), 503(100)	+		HMDB03213
**Phenolic acids**										
**5**	4.58	295.0667	295.0670	−1.02	C_10_H_16_O_10_	Malic acid hexoside	115(100), 133(52)		+	-
**6**	6.57	295.0667	295.0670	−1.02	C_10_H_16_O_10_	Malic acid hexoside isomer	115(100), 133(53)		+	-
**7**	11.47	169.0149	169.0142	4.14	C_7_H_6_O_5_	Gallic acid	125(100), 169(15)		+	[31]
**8**	15.59	153.0211	153.0193	11.76	C_7_H_6_O_4_	Protocatechuic acid	109(100)/153(35)		+	HMDB01856
**9**	17.04	285.0611	285.0615	−1.40	C_12_H_14_O_8_	Dihydroxybenzoic acid pentoside	108(100)/153(23)	+		-
**10**	21.48	417.1038	417.1038	0.00	C_17_H_22_O_12_	Dihydroxybenzoic acid dipentoside	108(21), 152(41), 241(5), 285(2), 417(100)	+		-
**11**	27.57	299.0771	299.0772	−0.33	C_13_H_16_O_8_	Hexosyl-oxy-benzoic acid	Level 3	+		-
**Amino acids**										
**12**	13.00	164.0712	164.0717	−3.05	C_9_H_11_NO_2_	Phenylalanine	72(1), 103(8), 147(29), 164(100)	+	+	[32]
**Flavone-*C*-glycosides**										
**13**	29.39	609.1459	609.1461	−0.33	C_27_H_30_O_16_	Luteolin 6-*C* hexoside 8-*C*-arabinoside	369(100), 399(41), 489(49), 609(91)	+		HMDB29258
**14**	31.63	593.1508	593.1511	−0.51	C_27_H_30_O_15_	Apigenin-6,8-*C*-dihexoside (vicenin-2)	353(52), 383(34), 473(31), 593(100)	+	+	HMDB30708
**15**	34.36	447.0931	447.0932	−0.22	C_21_H_20_O_11_	Luteolin-8-*C*-glucoside (orientin)	133(5), 285(9), 299(17), 327(100), 357(29), 447(44)	+		HMDB30614
**16**	34.38	563.1399	563.1406	−1.24	C_26_H_28_O_14_	Apigenin 6-*C*-hexoside-8-*C* pentoside I	297(17), 353(41), 383(28), 443(30), 473(20), 503(3), 545(5), 563(100)	+		FDB004202
**17**	34.71	447.0926	447.0932	−1.34	C_21_H_20_O_11_	Luteolin-6-C-glucoside (isoorientin)	133(9), 285(20), 297(32), 327(100), 357(69), 447(56)	+	+	FDB012395
**18**	35.81	431.0975	431.0983	−1.86	C_21_H_20_O_10_	Apigenin-8-*C*-glucoside (vitexin)	283(56), 311(100), 341(9), 431(58)	+	+	[33]
**19**	36.18	563.1400	563.1406	−1.07	C_26_H_28_O_14_	Apigenin 6-*C*-hexoside-8-*C* pentoside II	297(15), 353(39), 383(26), 443(31), 473(23), 503(2), 545(4), 563(100)	+		FDB000137
**20**	36.95	431.0975	431.0983	−1.86	C_21_H_20_O_10_	Apigenin-6-*C*-glucoside (isovitexin)	269(9), 283(55), 311(100), 341(37), 353(4), 431(54)	+	+	[33]
**21**	38.42	577.1561	577.1562	−0.17	C_27_H_30_O_14_	Rhoifolin (apigenin-7-*O*-neohesperidoside)	269(100), 577(20)	+		HMDB38848
**22**	39.98	269.0449	269.0455	−2.23	C_15_H_10_O_5_	Apigenin	107(2), 117(11), 151(4), 269(100)	+	+	[34]
**23**	40.86	253.0498	253.0506	−3.16	C_15_H_10_O_4_	7,4’-Dihydroxyflavone	117(15), 135(7), 253(100)		+	HMDB0247290
**24**	41.20	285.0399	285.0404	−1.75	C_15_H_10_O_6_	Luteolin	133(29), 151(5), 175(7), 285(100)	+	+	[30]
**25**	41.75	285.0388	285.0404	−5.61	C_15_H_10_O_6_	Fisetin	135(27), 163(15), 285(100)	+	+	[35]
**Flavonols**										
**26**	36.71	593.1505	593.1511	−1.01	C_27_H_30_O_15_	Kaempferol-3-*O*-rutinoside	151(1), 284, 285(100), 327(2), 593(71)	+	+	[30]
**27**	37.18	447.0931	447.0932	−0.22	C_21_H_20_O_11_	Kaempferol 7-*O*-glucoside	285(100), 327(3), 447(51)		+	HMDB0303599
**28**	38.00	577.1551	577.1562	−1.91	C_27_H_30_O_14_	Kaempferitrin	285(100), 431(4), 577(20)	+		[36]
**29**	41.30	345.0608	345.0615	−2.03	C_17_H_14_O_8_	Limocitrin	330(96), 345(100)		+	[37]
**Isoflavones**										
**30**	40.86	253.0509	253.0506	1.19	C_15_H_10_O_4_	Daidzin	Level 3		+	HMDB33991
**Pentacyclic triterpenoids**										
**31**	40.00	425.3745	425.3788	−10.11	C_30_H_50_O	Lupeol	Level 3	+	+	-
**Fatty acids**										
**32**	40.90	275.2011	275.2016	−1.82	C_18_H_28_O_2_	Stearidonic acid	Level 3	+	+	-
**33**	41.36	255.2331	255.2329	0.78	C_16_H_32_O_2_	Palmitic acid	Level 3		+	-
**34**	41.42	277.2161	277.2173	−4.33	C_18_H_30_O_2_	Linolenic acid	Level 3	+	+	-
**35**	42.20	327.2304	327.2329	−7.64	C_22_H_32_O_2_	DHA	Level 3	+		-
**36**	43.51	281.2519	281.2486	11.73	C_18_H_34_O_2_	Oleic acid	Level 3		+	-
**37**	43.71	585.4850	585.4888	−6.49	C_38_H_66_O_4_	FAHFA 38:4	Level 3		+	-
**38**	43.81	279.2333	279.2329	1.43	C_18_H_32_O_2_	Linoleic acid	Level 3	+		-
**39**	44.29	363.2509	363.2540	−8.53	C_22_H_36_O_4_	FAHFA 22:3	Level 3		+	-
**40**	44.83	293.2116	293.2122	−2.05	C_18_H_30_O_3_	Hydroxy octadecatrienoic acid	221(13), 227(25), 277(65), 293(100)	+		[38]
**Miscellaneous**										
**41**	28.35	401.1446	401.1453	−1.75	C_18_H_26_O_10_	Benzyl alcohol-hexoside-pentoside	89(20), 101(52), 133(70), 193(41), 233(26), 269(61), 401(100)	+		HMDB41514
**42**	32.11	415.1578	415.1609	−7.47	C_19_H_28_O_10_	Benzyl-*O*-rutinoside	Level 3	+		-
**Unknowns**										
**43**	3.36	201.0250	201.0252	−0.99	C_4_H_10_O_9_	Unknown	-	+	+	-
**44**	3.76	317.0514	317.0514	0.00	C_12_H_14_O_10_	Unknown	-	+		-
**45**	22.43	162.0555	162.0560	−3.09	C_9_H_9_NO_2_	Unknown alkaloid	-		+	-
**46**	29.24	385.1861	385.1867	−1.56	C_19_H_30_O_8_	Unknown	-	+		-
**47**	29.27	445.2076	445.2079	−0.67	C_21_H_34_O_10_	Unknown	-	+		-
**48**	38.52	607.1669	607.1668	0.16	C_28_H_32_O_15_	Unknown	-	+	+	-
**49**	39.29	431.2268	431.2286	−4.17	C_21_H_36_O_9_	Unknown	-	+		-

^1^ The “+” symbol indicates presence in the analyzed samples (leaf and/or flower).

**Table 3 plants-13-00976-t003:** Molecular docking scores of important metabolites of *Colvillea racemosa* on different protein targets involved in lung cancer.

Compound	Target Biomolecule	Docking Score (kcal/mol)	RMSD	H-Bonding Interactions with Active Site Amino Acids
QYT	γ PI3K (PDB ID: 2A5U)	−6.6005	0.3195	Asp 964, Val 882, Ile 881, Lys 833, Ser 806
Vitexin		−7.5173	1.4669	Asp 964, Asp 950, Val 882, Ile 879, Ser 806, THR 887
Kaempferol rutinoside		−8.7140	1.8553	Asp 964, Asp 950, Lys 890, Val 882
Luteolin		−6.3425	08061	Asp 841, Asp 964, Ile 963
Isoorinetin		−7.1223	1.3817	Asp 950, Asp 964, Lys 807, Lys 833
AQ4	EGFR (PDB ID: 1M17)	−8.5478	1.6563	Thr 766, Met 769
Vitexin		−7.3955	1.2149	Met 742, Val 702, Asp 831
Kaempferol rutinoside		−9.3078	2.3618	Ala 719, Asp 831, Arg 817, Asn 818
Luteolin		−6.2791	1.0875	Met 769, Glu 738, Met 742
Isoorinetin		−8.0587	1.5661	Met 742, Asp 831, Leu 694, Met 769
DTQ	CDK2 (PDB ID: 1DI8)	−7.1365	0.4501	Glu 81, Asp 145, Leu 83, Phe 80
Vitexin		−7.5700	1.5588	Gln 131, Gly 11, Phe 82, Leu 83, Val 18
Kaempferol rutinoside		−8.5236	3.3823	Asp 145, Val 18, Ala 144
Luteolin		−6.3872	0.6960	Leu 83, Asp 145, Val 18, Ala 144
Isoorinetin		−6.8739	3.5786	Asp 145, Lys 89, Val 18

## Data Availability

Data will be made available upon request.

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
