# Peer review of "Metabolite Profiling of Colvillea racemosa via UPLC-ESI-QTOF-MS Analysis in Correlation to the In Vitro Antioxidant and Cytotoxic Potential against A549 Non-Small Cell Lung Cancer Cell Line"

_plants, 2024, doi:10.3390/plants13070976_

Round 1

Reviewer 1 Report

Comments and Suggestions for Authors

The study presented is interesting and worth publishing in Plants with additional analysis. The methodological approach is scientific. The results clearly shows the the anticancer or cytotoxic effect of the leaf and flower extracts against A 549 lung cancer cell line.

The authors should consider testing on one or two more lung cancer cell lines

There is no proper mechanistic investigation (i.e. expression studies) presented. Hence, I recommend the authors to probe many other proteins (using computational docking) that are closely linked to cancer and apoptosis; not just PI3K This will help the authors to speculate better about the possible anticancer mechanisms.

ROS production may be assessed in the cells and linked to the in vitro antioxidant activity.

Comments on the Quality of English Language

Good

Author Response

The study presented is interesting and worth publishing in Plants with additional analysis. The methodological approach is scientific. The results clearly show the anticancer or cytotoxic effect of the leaf and flower extracts against A 549 lung cancer cell line.

Response: We are thankful for the reviewer for his comments. The required improvements have been made accordingly. We hope that our revised manuscript will be accepted for further publication in “Plants”.

  1. The authors should consider testing on one or two more lung cancer cell lines

Response: we thank the reviewer for his comment. Our focus on this work is mainly non-small cell lung cancer (NSCLC) which is represented by A549 cell lines. Accordingly, the title of the manuscript was updated as follows:

Metabolite profiling of Colvillea racemosa via UPLC-ESI-QTOF-MS analysis in correlation to the in vitro antioxidant and cytotoxic potential against A549 Non-Small Cell Lung Cancer cell lines

Also a paragraph was added to highlight this type of lung cancer which is the most predominant one.

Lines 92-97: Lung cancer can be divided into small cell lung cancer (SCLC) and non-small cell lung cancer (NSCLC) groups [1]. NSCLC constitutes approximately 85% of all lung cancer cases [2,3]. Notably, surgery is the preferred and main approach for early-stage NSCLC, which offers a significant chance for long-term survival for those patients [4,5]. However, late stages require platinum-based combinations as chemotherapeutics [6].

  1. There is no proper mechanistic investigation (i.e. expression studies) presented. Hence, I recommend the authors to probe many other proteins (using computational docking) that are closely linked to cancer and apoptosis; not just PI3K This will help the authors to speculate better about the possible anticancer mechanisms.

Response: Docking was conducted for two other hallmark biotargets in colorectal cancer, EGFR and CDK2. Our tested flavonoids showed very favorable interactions with both targets being at some instances better binders than the co-crystallized ligands. Updated data can be found in Table 3 and figures 5-7. In addition, the discussion of the molecular docking studies was improved and extended accordingly.

Lines 322-347: Additionally, EGFR is another important target in cancer management for cancers associated with overexpression of this receptor tyrosine kinase like colorectal cancer. This prompted us to invetigate the binding interaction potential of the most abundant flavonoids viz; vitexin, kaempferol rutinoside, luteolin and isoorientin with EGFR. Table 3 and Fig. 5 display the docking findings of the examined compounds in comparison to the co-crystallized ligand AQ4 (S-score -8.5478 kcal/mol). Results showed that all four compounds demonstrated good recognition of the active site amino acids with S-scores of -7.3955, -9.3078, -6.2791 and -8.0587 Kcal/mol, respectively. kaempferol rutinoside showed better binding than the co-crystallized ligand as reflected by its S-score forming a network of H-bonds with Ala 719, Asp 831, Arg 817 and Asn 818 active site amino acids.

Furthermore, CDK2 is also considered an important protein target in cancer [7]. As such, we wanted to determine the potential effect of the selected flavonoids viz; vitexin, kaempferol rutinoside, luteolin and isoorientin on CDK2. In silico investigation results reflected the ability of these flavonoids to recognize and interact with the key amino acids of the active site of CDK2 in a comparable manner to the co-crystallized inhibitor DTQ (Table 3 and Fig. 6). While the co-crystallized ligand displayed and S-score for binding of -7.1365 kcal/mol, the investigated flavonoids had better or near S-scores of -7.5700, -8.5236, -6.3872 and -6.8739 kcal/mol, respectively. The best binder again was kaempferol rutinoside forming two H-bonds with both Asp 145 and Val 18, in addition to a pi-H interaction with Ala 144.

Details of the 3D interactions of the most active compound in the docking study by virtue of its high binding S-scores, kaempferol rutinoside, are presented in Fig. 7. With all 3 biological targets, PI3K, EGFR and CDK2, kaempferol rutinoside was able to establish a network of H bonds and pi-H bonding interactions with the active site amino acids to stabilize its complex with these biotargets and hence elicit its inhibitory activity paving the way to the anticancer effect of C. racemosa it participates to.

  1. ROS production may be assessed in the cells and linked to the in vitro antioxidant activity.

Response:

We thank the reviewer for his comment. Unfortunately, we could not assess the ROS production in the cells because of the limited time of the revision period for our manuscript. This investigation will be considered in our future work on Colvillea racemosa extracts for further evaluation of their in vivo efficacy on lung cancer models.

Reviewer 2 Report

Comments and Suggestions for Authors

In this study, the authors evaluated the bioactive compounds from flower and leaf extracts of Colvillea racemosa using UPLC-ESI-QTOF-MS, and the bioactivity was tested using in vitro antioxidant assays (FRAP, DPPH, and ABTS) and cytotoxic assays on lung cancer cell lines (A-549). The methodology and writing are clear and should provide interesting information for readers. However, I believe that further improvement is necessary before publication. My comments are as follows.

1. Plant material should be accompanied by voucher specimen numbers, as it is crucial for sample identification. Additionally, I suggest providing the coordinates for the sampling locations in Egypt.

2. Furthermore, I recommend including a reference column for each metabolite in Table 2. Additionally, for the MS/MS column, I request the inclusion of peaks along with their respective fragmentations.

3. The docking section requires further discussion, including the incorporation of 3D figures and important connections beyond hydrogen bonding interactions with amino acids.

4. Lastly, the Conclusion section needs to be rewritten in more depth and with appropriate citations.

Comments on the Quality of English Language

Moderate editing of English language required

Author Response

In this study, the authors evaluated the bioactive compounds from flower and leaf extracts of Colvillea racemosa using UPLC-ESI-QTOF-MS, and the bioactivity was tested using in vitro antioxidant assays (FRAP, DPPH, and ABTS) and cytotoxic assays on lung cancer cell lines (A-549). The methodology and writing are clear and should provide interesting information for readers. However, I believe that further improvement is necessary before publication. My comments are as follows.

Response: We are very grateful for the reviewer for his comments. The required improvements have been made accordingly. We hope that our revised manuscript will be accepted for further publication in “Plants”.

  1. Plant material should be accompanied by voucher specimen numbers, as it is crucial for sample identification. Additionally, I suggest providing the coordinates for the sampling locations in Egypt.

Response: Voucher specimen number and coordinates of the sampling locations are now added in the text as follows:

Lines 385-392: Leaves and flowers of C. racemosa Bojer ex Hook were collected in July 2019 from Mostafa EL-Abd Gardens, Cairo-Alexandria Desert Road, Egypt (30.3552330 N, 30.5166490 W). The plant material was kindly identified by Mrs. Therese Labib, Consultant of Plant Taxonomy at the Ministry of Agriculture and Orman Botanical Garden, Giza, Egypt. Voucher specimen (No. 10.7.2019) was kept in the herbarium of Department of Pharmacognosy, Faculty of Pharmacy, Cairo University.

  1. Furthermore, I recommend including a reference column for each metabolite in Table 2. Additionally, for the MS/MS column, I request the inclusion of peaks along with their respective fragmentations.

Response: A reference column was added for the metabolites in the revised manuscript and also the peak intensities of the MS/MS fragments.

  1. The docking section requires further discussion, including the incorporation of 3D figures and important connections beyond hydrogen bonding interactions with amino acids.

Response: 3D figure representing the interactions of the most potent kaempferol rutinoside was added, and the discussion was improved accordingly.

Lines 342-350: Details of the 3D interactions of the most active compound in the docking study by virtue of its high binding S-scores, kaempferol rutinoside, are presented in Fig. 7. With all 3 biological targets, PI3K, EGFR and CDK2, kaempferol rutinoside was able to establish a network of H bonds and pi-H bonding interactions with the active site amino acids to stabilize its complex with these biotargets and hence elicit its inhibitory activity paving the way to the anticancer effect of C. racemosa it participates to.

  1. Lastly, the Conclusion section needs to be rewritten in more depth and with appropriate citations.

Response: Thanks for the suggestion. Based on the new results obtained in the docking models, we have improved the conclusions by making them more in-depth and highlighting the potential bioactive effect of C. Racemose extract and their specific metabolites.

Moderate editing of English language required

Response: The English language was revised and edited accordingly.

Reviewer 3 Report

Comments and Suggestions for Authors

In the article entitled “Metabolite profiling of Colvillea racemosa via UPLC-ESI- QTOF-MS analysis in correlation to the in vitro antioxidant and cytotoxic potential”. The authors provided interesting research, highlighting the antioxidant and cytotoxic activities of metabolites in Colvillea racemose leaf and flower extracts obtained through UPLC-ESI-QTOF-MS. Nevertheless, there are some suggestions to further improve the manuscript:

1-       On page 2 line 52, the authors should remove the phrase “hey exhibited multiple pharmacological activities viz”. Because it repeats the same meaning as the previous phrase.

2-       In the introduction, the antioxidant and cytotoxic effects of the Fabaceae family are briefly mentioned. It is highly recommended to extend this part by introducing more recent works.

3-       In the discussion, the authors should mention the limitations of their work and how they can improve it in their future works.

4-       On page 8, line 215, the authors should remove the phrase “calcium overload”. Because it is repeated twice.

5-       The authors must improve the resolution of figures 3 and 4 because it is poor.

6-       The manuscript required moderate editing of the English language. The authors should revise it and correct it.  

7-       p. 1, line 19: The term "as a source" suggests that there must be "a source".

8-       p. 1, line 22: The term "bioactivity were" suggests that there must be "bioactivity was".

9-       p. 1, line 32: The term "study" suggests that there must be "studies".

10-  p. 2, line 64: The term "holding" suggests that there must be "which holds".

11-  p. 2, line 72: The term "there is" suggests that there must be "there has been".

12-  p. 2, line 73: The term "reducing" suggests that there must be "reduce".

13-  p. 2, line 74: The term "decreasing" suggests that there must be "decrease".

14-  p. 2, line 90: The phrase "to suppress the harmful effect" suggests that there must be a "to suppressing the harmful effects".

15-  p. 2 line 92: The term "assay" suggests that there must be "assays".

16-  p. 3, line 113: The term "comparing" suggests that there must be "compared ".

17-  p. 6, line 125: The term "either aglycones" suggests that there must be "either as aglycones".

18-  p. 6, line 165: The term "showing" suggests that there must be "showed".

19-  p. 7, line 186: The term "as well as" suggests that there must be "and".

20-  p. 7, line 198: The term "suggestive of" suggests that there must be "suggesting of".

21-  p. 8, line 214: The term "induced" suggests that there must be "induces".

22-  p. 9, line 249: The term "revealed as" suggests that there must be "revealed to be".

23-  p. 9, line 250: The term "induction" suggests that there must be "inducing".

24-  p. 9, line 253: The term "promissing " suggests that there must be "promising".

25-  p. 10, line 270: The term "till" suggests that there must be "until".

26-  P. 11, line 322: The term "under darkness" suggests that there must be "in darkness".

27-  p. 11, line 328: The term "of Boly" suggests that there must be "by Boly".

28-  p. 11, line 341: The term "persulphate" suggests that there must be "persulfate".

29-  p. 12, line 375: The term "with its binding site" suggests that there must be "and their binding sites".

Comments on the Quality of English Language

In the manuscript entitled “Metabolite profiling of Colvillea racemosa via UPLC-ESI- QTOF-MS analysis in correlation to the in vitro antioxidant and cytotoxic potential”. The authors have written the article in comprehensive English with moderate grammatical errors.

Author Response

In the article entitled “Metabolite profiling of Colvillea racemosa via UPLC-ESI- QTOF-MS analysis in correlation to the in vitro antioxidant and cytotoxic potential”. The authors provided interesting research, highlighting the antioxidant and cytotoxic activities of metabolites in Colvillea racemose leaf and flower extracts obtained through UPLC-ESI-QTOF-MS. Nevertheless, there are some suggestions to further improve the manuscript:

Response: We are thankful for the reviewer for his comments. The required improvements have been made accordingly. We hope that our revised manuscript will be accepted for further publication in “Plants”.

1-       On page 2 line 52, the authors should remove the phrase “hey exhibited multiple pharmacological activities viz”. Because it repeats the same meaning as the previous phrase.

Response: Done.

2-       In the introduction, the antioxidant and cytotoxic effects of the Fabaceae family are briefly mentioned. It is highly recommended to extend this part by introducing more recent works.

Response: New paragraphs were added with recent work regarding the antioxidant and cytotoxic effects of the Fabaceae family

Lines 48-61: Many studies reported the antioxidant activity of different plants of this family. For instance, Gođevac et al.[3], studied nine Fabaceae species growing in Serbia and Montenegro, where Lathyrus binatus, Trifolium pannonicum, and Anthyllis aurea revealed as the most potent antioxidant plants using different antioxidant assays. Additionally, the seed extract of Mucuna pruriens provoked antioxidant effects using DPPH, FRAP and ABTS assays. The results were comparable to those of rutin and gallic acid used as standards [4].  

Moreover, family Fabaceae has been studied for cytotoxic potential. An alkaloidal fraction prepared from Indigofera suffruticosa (a member of family Fabaceae) exhibited cytotoxic activity with IC50 values of 37.39 and 24.13 µg/ml against LM2 (breast adenocarcinoma) and LP07 (lung adenocarcinoma) cell lines, respectively [5]. Also, the extract of Senna singueana at 100 mg/kg showed in vivo antiapoptotic effect on liver cells by reduction of Bcl-2 [6]. Importantly, other studies revealed low cytotoxicity of other plants belonging to Fabaceae on Vero cell line [7].

3-       In the discussion, the authors should mention the limitations of their work and how they can improve it in their future works.

Response:

Lines 350-358: On the other hand, aside from our in vitro findings on the cytotoxic potential of C. racemosa extracts, extrapolation on in vivo testing will be our future perspective. Also, detailed risk assessment and systemic toxicity profile should be investigated to guard against any cytotoxic effects on normal tissues. Additionally, after sufficient in vivo exploration, the synergistic effect of C. racemosa extracts to the drugs used for NSCLC will be of great value to decrease the side effects of those synthetic drugs.

4-       On page 8, line 215, the authors should remove the phrase “calcium overload”. Because it is repeated twice.

 Response: Corrected.

5-       The authors must improve the resolution of figures 3 and 4 because it is poor.

Response: The resolution of figures 3 and 4 was improved accordingly.

6-       The manuscript required moderate editing of the English language. The authors should revise it and correct it.  

Response: The English language was revised and edited accordingly.

7-       p. 1, line 19: The term "as a source" suggests that there must be "a source".

 Response: Corrected.

8-       p. 1, line 22: The term "bioactivity were" suggests that there must be "bioactivity was".

 Response: Corrected.

9-       p. 1, line 32: The term "study" suggests that there must be "studies".

 Response: Corrected.

10-  p. 2, line 64: The term "holding" suggests that there must be "which holds".

 Response: Corrected.

11-  p. 2, line 72: The term "there is" suggests that there must be "there has been".

 Response:  Corrected.

12-  p. 2, line 73: The term "reducing" suggests that there must be "reduce".

 Response: Corrected.

13-  p. 2, line 74: The term "decreasing" suggests that there must be "decrease".

 Response: Corrected.

14-  p. 2, line 90: The phrase "to suppress the harmful effect" suggests that there must be a "to suppressing the harmful effects".

 Response: Corrected.

15-  p. 2 line 92: The term "assay" suggests that there must be "assays".

 Response: Corrected.

16-  p. 3, line 113: The term "comparing" suggests that there must be "compared ".

 Response: Corrected.

17-  p. 6, line 125: The term "either aglycones" suggests that there must be "either as aglycones".

 Response: Corrected.

18-  p. 6, line 165: The term "showing" suggests that there must be "showed".

 Response: Corrected.

19-  p. 7, line 186: The term "as well as" suggests that there must be "and".

 Response: Corrected.

20-  p. 7, line 198: The term "suggestive of" suggests that there must be "suggesting of".

 Response: Corrected.

21-  p. 8, line 214: The term "induced" suggests that there must be "induces".

 Response: Corrected.

22-  p. 9, line 249: The term "revealed as" suggests that there must be "revealed to be".

 Response: Corrected.

23-  p. 9, line 250: The term "induction" suggests that there must be "inducing".

 Response: Corrected.

24-  p. 9, line 253: The term "promissing " suggests that there must be "promising".

 Response: Corrected.

25-  p. 10, line 270: The term "till" suggests that there must be "until".

 Response: Corrected.

26-  P. 11, line 322: The term "under darkness" suggests that there must be "in darkness".

 Response: Corrected.

27-  p. 11, line 328: The term "of Boly" suggests that there must be "by Boly".

 Response: Corrected.

28-  p. 11, line 341: The term "persulphate" suggests that there must be "persulfate".

 Response: Corrected.

29-  p. 12, line 375: The term "with its binding site" suggests that there must be "and their binding sites".

 Response: Corrected.

In the manuscript entitled “Metabolite profiling of Colvillea racemosa via UPLC-ESI- QTOF-MS analysis in correlation to the in vitro antioxidant and cytotoxic potential”. The authors have written the article in comprehensive English with moderate grammatical errors.

Response: The English language was revised and edited accordingly.

Round 2

Reviewer 1 Report

Comments and Suggestions for Authors

I am ok with the authors' responses and the revisions done on the manuscript

Reviewer 2 Report

Comments and Suggestions for Authors

The authors have significantly improved the revised manuscript following my suggestion. Therefore, I am satisfied to accept this manuscript.

Comments on the Quality of English Language

Minor editing of English language required